# Bioinspired Pattern-Driven Single-Material 4D Printing for Self-Morphing Actuators

**Yousif Saad Alshebly** [1], **Khameel B. Mustapha** [2], **Ali Zolfagharian** [3], **Mahdi Bodaghi** [4], **Mohamed Sultan Mohamed Ali** [5], **Haider Abbas Almurib** [1] **and Marwan Nafea** [1,*]

1   Department of Electrical and Electronic Engineering, Faculty of Science and Engineering, University of Nottingham Malaysia, Semenyih 43500, Selangor, Malaysia
2   Department of Mechanical, Materials and Manufacturing Engineering, Faculty of Science and Engineering, University of Nottingham Malaysia, Semenyih 43500, Selangor, Malaysia
3   School of Engineering, Deakin University, Geelong, VIC 3216, Australia
4   Department of Engineering, School of Science and Technology, Nottingham Trent University, Nottingham NG11 8NS, UK
5   School of Electrical Engineering, Faculty of Engineering, Universiti Teknologi Malaysia (UTM), Johor Bahru 81310, Johor, Malaysia
*   Correspondence: marwan.nafea@nottingham.edu.my

**Abstract:** Four-dimensional (4D) printing of shape memory polymers is a leading research field due to the possibilities allowed by using these materials. The strain difference in the structures that is caused by the different stiffness profiles can be used to influence the shape-memory effect in the actuators. In this study, the influence of patterns on the strain is tested in polylactic acid (PLA) actuators using patterns made of different shapes. Five bioinspired geometrical shapes, namely, circles, squares, hexagons, rhombuses, and triangles, are used in the three-dimensional (3D) printing of the actuators. The use of shapes of different sizes along with combinations of different patterns in the PLA actuators is carried out to develop 40 actuators with different designs. The effects of the patterns and their characteristics are analysed and compared. The self-bending angles of the actuators range from 6.19° to 30.86°, depending on the patterns and arrangement used. To demonstrate the feasibility of utilizing the proposed designs in practical applications, a hand-like shaped gripper is developed. The results show that the gripper can grip objects with uniform and non-uniform cross-sections. The developed gripper demonstrates that the proposed concept can be implemented in various applications, including self-morphing structures and soft robotics.

**Keywords:** 4D printing; bioinspired; pattern-driven; polylactic acid; shape memory polymer

## 1. Introduction

Four-dimensional (4D) printing is a rapidly improving category of additive manufacturing (AM). This category combines the control that is achievable in AM with the favourable characteristics of smart materials. Many types of smart materials are used in 4D printing, varying from shape memory, to colour changing or self-healing materials. The variety in materials means that the methods of activation of the printed structures can be tailored to specific applications, where the use of certain triggers is not possible. Coupled with many AM techniques, 4D printing has the potential to improve many fields, such as soft robotics, biomedical engineering, and micro-actuators [1]. The field has been advancing at its current rate because of the freedom it allows in the manufacturing process [2]. The quick development of functional prototypes, combined with the degree of freedom and the ease of complexity, allowed the fast generation of improvements in the fields of biomedicine [3], construction [4], and rapid complex prototyping [5]. The field is supported by a plethora of advancements in material science, allowing the use of materials with favourable characteristics alongside the sophistication of prototyping in AM [6]. Certain

aspects of 4D printing can make structures adaptive to their environment if their stimuli are environmental. This characteristic makes them suitable as passive sensors [7]. The shape-changing effect has been used in self-assembling or activating structures, such as hybrid hinges structures [8] and bio-medical implants [9].

Shape memory polymers (SMPs) are a leading type of material used in 4D printing due to their ability to store a high amount of energy as strain, coupled with the highly versatile ability to change their shapes as a release of that strain [10]. Popular SMPs in this field, such as polyurethane-based polymer [11] and polylactic acid (PLA) [12], have good shape memory characteristics. In addition, these SMPs offer comparatively low costs, allowing them to be widely made into composites. These composites are used to change the shape memory effect (SME) and the deformation of the polymers. Other composites are created in order to control the activation of the stimulus for various applications [13]. PLA, which is a biodegradable thermoplastic based on lactic acid that can be produced from renewable resources, such as corn and sugar beets, is easy to use, making it the preferred material for 4D printing and biomedical researchers [14,15].

The shape change of the materials is triggered by using external stimuli, such as temperature [8], solvents [16], pH [17], magnetics [18], and light [19]. This allows the materials to release the internal strain that holds them in a temporary shape. Most SMPs are activated via heating, where they transition from a glassy to a rubbery state, making them easily mendable into many temporary shapes [20]. The use of heat, or a heated liquid environment, is the simplest method of activation because it requires no additional additives to the materials. Moreover, such approaches allow full and equal heating of the structure from all sides [21]. In manual shape programming, force is applied to the heated material to mend it to its temporary shapes. The material is then cooled below its transition temperature, causing it to hold its shape with a pre-strain. The structure goes back to its original shape when it is heated again beyond the glass transition temperature [7]. This approach offers the ability to develop complex structures that utilize different actuation mechanisms, which is a challenging task to be achieved when using other types of actuation methods that rely on complex fabrication steps [22,23]. There are many AM printing methods used in 4D printing, with the most popular being fused deposition modelling (FDM). FDM uses thermoplastics as filament for the prints, which allows vast control over the printing method [24]. In FDM, the material is extruded through a heated nozzle one line at a time, creating a model. FDM is the most popular method because of the relatively low cost of printing and the freedom of use in terms of materials since it uses filaments to print [25].

Based on the way they are developed, 4D-printed actuators can undergo shape changes in many forms that range from bending to twisting and curving. Many of these can be implemented into structures capable of complex shape changes [26]. These changes in shape are traditionally allowed by training the materials, which is the process of using force to induce strain on the material. Another method of inducing strain in the materials is by controlling the printing parameters [27], which allows the training of the materials without the use of force after printing. By controlling the printing parameters, the internal strain can be induced at high levels of control, especially in FDM [28]. Many researchers prefer to use these methods to train SMPs because they allow them to obtain a strain that is ready to use when they receive a stimulus. As a proof of concept, grippers are widely used to test the usability of actuators in various applications due to their common use in soft robotics [29]. The use of 4D printing for robotic actuators provides advantages over mechanical robotics since they can grasp objects without comprising any joints, allowing flexible and adaptive gripping mechanisms [30]. There are several basic design principles for such a structure design, such as bistable and multi-stable designs [31] and bi-layer designs that utilise a combination of polymer-paper composite [32].

Bioinspired 4D-printed structures have emerged as desired designs in many application-oriented systems. This concept has been a big part of multiple leading technological advancements because of the freedom it allows in the formation of structures, providing

the ability to make agile mechanical movements in a rigid body [33]. In addition, bioinspired designs facilitate topology optimisation, enable energy-efficient actuation, promote embodied intelligence, and allow flexible robots to exhibit different behaviour than traditional structures [34,35]. Bioinspired 4D printing has been applied in many fields due to its ability to control many characteristics in the structure that may be uncontrollable by materials preference or too complex and non-compliant for traditional mechanisms. The structures of various living organisms have been mimicked by many researchers to develop 4D-printed bioinspired flexible and adaptable devices [36]. For instance, Faber et al. utilised programmable folding based on the wing of the *Forficula auricularia* to 4D print a structure using PLA and thermoplastic polyurethane [37]. In another work, Liu et al. developed 4D-printed hydrogel tubes that were inspired by the structure of the coral polyp [38]. Bioinspired designs were also investigated by McCracken et al., who used ionic hydrogels to 4D print biomimetic aquatic organisms [39], and Gladman et al., who developed biomimicking plants for complex planar to 3D morphology [40]. Another research team developed leaf-like structures that were inspired by the reversible shape-changes of *Pinus wallichiana*, which were 4D printed using a wood-polymer composite and acrylonitrile butadiene styrene (ABS) [41]. Peng et al. developed biomimetic leaf, snail shell, and worm-like structures that were 4D printed using PLA [42]. Other researchers utilised the tensioning of the stem helix of the *Dioscorea bulbifera* to develop a 4D-printed self-tightening splint [43]. The structure was printed using a combination of wood-polymer composite and ABS. In another research, Zeng et al. proposed bioinspired 4D-printed actuators based on the adductor muscle and hinge of the muscle-shell using PLA [44].

Despite the promising advancement achieved in bioinspired 4D printing, there are still many areas that require investigation and improvements. The possibilities of living organisms that can be used as inspiration for biomimetics are unlimited. In addition, the effect of printing patterns on the performance of 4D-printed actuators has not been explored well so far. Motivated by the aforementioned promising features of bioinspired 4D-printed structures and the limitations that need to be addressed in this area, this work presents single-material bioinspired actuators that offer different actuation performances based on the printing patterns used. This work uses the induced strain that is achievable in the three-dimensional (3D) printing method of FDM and controls it using the printing patterns. The stiffness of materials can be manipulated in AM by printing the materials in patterns, giving different stiffness profiles to the structures. The main contributions of this paper to the reported literature are:

- This study investigates the effect of five bioinspired printing patterns on the actuation performance of 4D-printed actuators, which are shown in Figure 1a–e. These shapes are inspired by *Volvox carteri* (circles), *Haloquadratum walsbyi* (squares), honeycombs (hexagons), *Sapium sebiferum* (rhombuses), and *Oxalis triangularis* (triangles).
- This study presents a novel approach of using geometrical patterns to create controllable strain differences between the layers of prints while assessing the effects of geometrical shape patterns on the actuation of 4D-printed actuators.
- The proposed concept is used to develop a hand-like shaped gripper that can grip and pick up delicate objects with uniform and non-uniform cross-sections. This is achieved by utilising the different actuation angles caused by the printing patterns, where the effects of full-structure patterns are not reported, to the best knowledge of the authors.

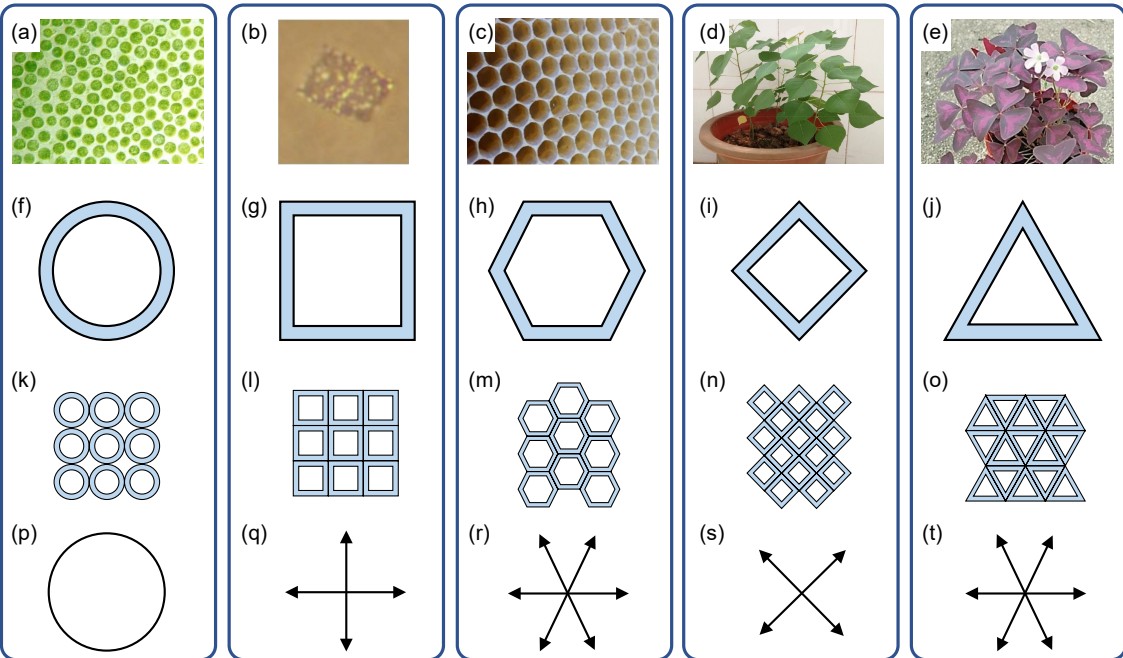

**Figure 1.** Bioinspired designs' concepts. (**a**) *Volvox carteri* [45]. (**b**) *Haloquadratum walsbyi* [46]. (**c**) Honeycomb [47]. (**d**) *Sapium sebiferum* adapted with permission from [48]. (**e**) *Oxalis triangularis* [49]. (**f**–**j**) Bioinspired shapes used to create the patterns of the actuators. (**k**–**o**) Bioinspired patterns created with each shape. (**p**–**t**) Overall pre-stress profiles of the printed patterns.

## 2. Design and Working Principle

An FDM printer was used for this study to print actuators using a PLA filament. The heating of the filament through the nozzle, coupled with the feed rate and stretching of the filament as it is being printed, adds strain to the material upon 3D printing. The strain inside the materials after printing is stored and will not be released until the material is heated above the glass transition temperature ($T_g$). When the strain is released from the material, the shape of the structure changes until a full release of the internal strain is achieved. This stored force is referenced as "pre-induced strain" in this work. The pre-strain is highly controllable [11]. The internal strain in the materials is influenced by the parameters set by the 3D printer, ranging from printing speed to printing temperature and other processing parameters. When the printing settings are maintained, the internal strain will remain the same and will have the same effect on the structure upon heating when repeated.

The patterns developed in this work have different stiffness in different directions. A direction stiffness profile can aid in the understanding of the SME caused by each pattern since the SME acts in the direction of printing, which is expanded and discussed in the next section. Different stiffness in certain directions will cause the SME to shape the structure in a certain way. This being the case, the mixing of patterns of different sizes and patterns will produce a different SME in the structure because of the difference in the stiffness of the two patterns. This variable stiffness can be useful in many SME programs through the use of patterns.

The material properties of PLA are tested using DSC and DMA. These properties are vital for the activation of the material to release all the internal strain. A DSC machine (DSC Q2000, Texas Instrument, Dallas, TX, USA) was used to generate the heat flow of the materials for heating and cooling, thus allowing the viewing of the effect of heating the material to activate the stimuli and the effect of the cooling to return the material to the glassy state. The test is conducted via heating from 30 °C to 250 °C at a rate of 10 °C per minute, then cooled down to 30 °C at the same rate using liquid nitrogen. In addition, a DMA machine (DMA 8000, Perkin Elmer, Waltham, MA, USA) was used to measure the

damping of the material (tan $\delta$), which is most needed in SMPs because it indicates the $T_g$ of the material, and the storage modulus ($E'$), which is used to assess the elasticity of the material at different temperatures. The test was conducted using axial tensile clamping at a frequency of 1 Hz. The heating of the material in the heating chamber was set to increase from 30 °C to 100 °C at a rate of 5 °C per minute.

### 2.1. Pattern Functionality

This section presents the main patterns used in this work along with their profiles. The five bioinspired shapes used in this work can be found in *Volvox carteri* (circles), *Haloquadratum walsbyi* (squares), honeycombs (hexagons), *Sapium sebiferum* (rhombuses), and *Oxalis triangularis* (triangles), as shown in Figure 1a–e. All the patterns are made by repeating a single shape for each pattern. In 3D printing, geometrical shapes are used in the infill, sometimes referred to as internal structures, and each changes the characteristics of the print in a certain way [50], in which basic geometrical shapes are used to create the infill patterns. Although there are many mechanical preferences in choosing these shapes, their effect on the SME is not explored. Each pattern has a different profile, referred to as a stiffness profile in this work. This profile plays a big role in the shape change experienced by the printed patterns when activated. The profiles of each pattern used in this work are demonstrated in Figure 1f–j. The deformation of the actuators can be realized by understanding the shape change in each geometrical formation used in the patterns. The printing of these patterns is done one shape at a time, which makes the SME work along the lines of each shape. The SME of each shape acts in a shrinking manner along the printing direction.

The main patterns have simple geometric shapes, where their SME can be analysed by their perimeters. Figure 1k–o show the patterns made by using each shape. To understand the shape change, a reverse directional motion of the actuation is represented in Figure 1p–t. This directional motion represents the pre-stress directions that are achieved during the printing process of the complete pattern. This can also be explained as the force that would have been applied to the patterns if it was done in the traditional programming way in order to achieve the induced-strain shape change. In this study, the printing process induces the strain, but the pre-stress profiles help explain the directions of the overall pre-stress. The force is parallel to the sides of the shapes of the patterns since the SME acts along with the perimeters in a shrinking manner. This shape profile is introduced in this study as an easier way to understand the deformations of the shapes since it has a centre and direction. Although the profiles in Figure 1r,t are the same, when placed in their patterns, they act differently since the shapes are in different relative positions to each other, meaning that each pattern will have a different force profile. Each circle shape in Figure 1k is pre-stressed equally in all directions across the print bed. Thus, there is no linear direction of pre-stress in Figure 1p.

With a more complex approach, further differences in the stiffness or force profiles in the patterns can be made. This can be done by having two different layers of the patterns, a bottom layer and a top layer. By using variations of the pattern for the bottom and top sections, different profiles can be created. In this study, there are three configurations of pattern profiles:

- The first configuration is simple patterns with a single shape for each pattern.
- The second configuration is made by dividing the actuators into two layers, where both layers use the same pattern, but the size and number of shapes used in each layer are different, creating a stronger profile difference between the two.
- The third configuration is made of actuators with two layers of mixed patterns, where each layer uses a different pattern.

### 2.2. Single Shape Patterns

The first set of actuators is the simplest one since the actuators are made of basic geometrical shapes next to each other, as demonstrated in Figure 1. These actuators are

the baseline ones, and the SME in them is small since their stiffness profiles are the same. Any shape change is mainly caused by the printing parameters. All actuators were printed with standard dimensions of 51.6 mm × 13.6 mm × 2 mm, representing length, width, and height, respectively. These dimensions make up the geometrical shapes that are enclosed by a perimeter. The sizing of the actuators without the perimeter was chosen to be 12 mm since it can hold 3 large shapes (4 mm each) or 4 small shapes (3 mm each), where a perimeter of 0.8 mm thickness was then added to the sides. The length was determined by repeating the shapes until they reached a standard length of 50 mm, while a 0.8 mm perimeter was added on each side. The height was set so that when dividing the actuators into two layers; each layer has a 1 mm height. The dimensions of all of the designs presented in this work can be found in detailed illustration in the Supplementary Materials.

### 2.3. Variable Pattern Sizes

The use of patterns of the same shape and different sizes for each layer of the actuators creates a bigger difference between the two layers, causing variable stiffness, especially at the boundary between the two different layers. Since the standard pattern uses shapes that are at a width of 4 mm for a total of 12 mm, the smaller patterns are made of 3 mm shapes, using 4 shapes for the width, for a total of 12 mm as well. The difference between the shape size between the bottom and top sections of the actuator generates a higher strain difference due to the variable stiffness in the actuator. The smaller shapes are not only smaller in height since the size change was made to keep the same formulation of the shape, creating a difference between the shapes in all directions. This difference changes the stiffness profile dramatically since the forces have different directions and centres, as demonstrated by the stiffness profile in Figure 2.

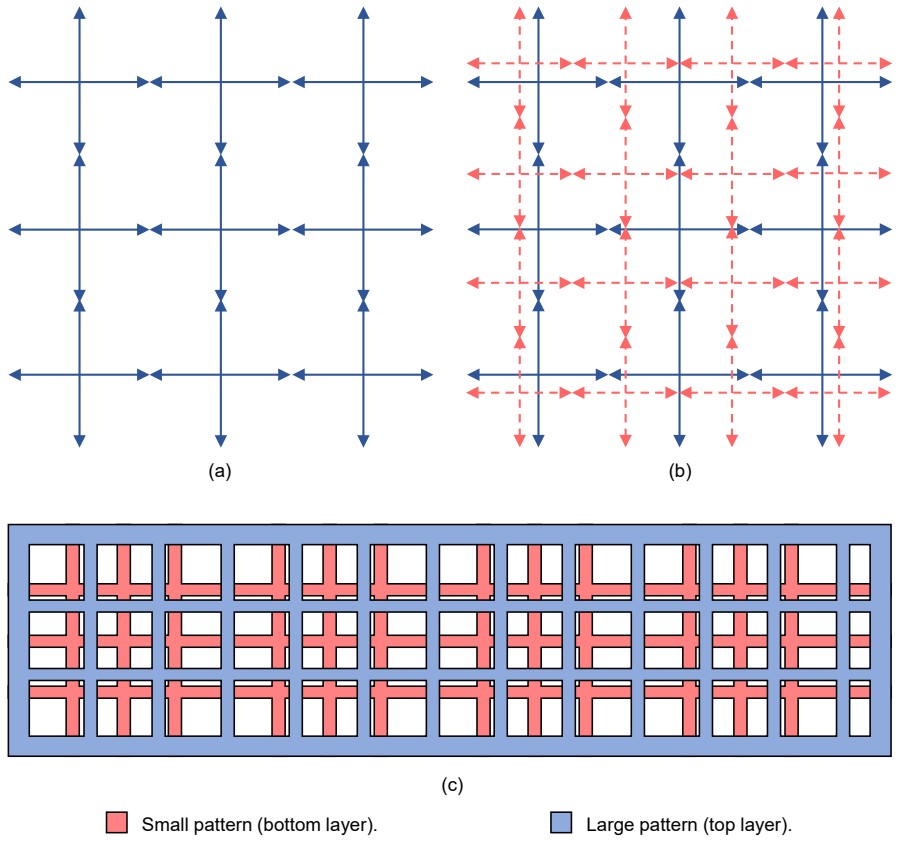

**Figure 2.** The difference in the stiffness, or the force profile, in the variable shape and size patterns. (**a**) The pre-strain profile of a square-pattern actuator with a single shape size. (**b**) The pre-strain profile of a square-pattern actuator with a variable shape size. (**c**) Top view of a square-pattern actuator with two layers of variable square sizes.

The shape presented in Figure 2a is the stiffness profile that is created in the pattern made by squares of the 4 mm size, which was described earlier. On the other hand, the shape presented in Figure 2b is the stiffness profile of an actuator that is made with small and large square patterns, placed one above the other. The blue-coloured arrows represent the profile of the large size square pattern, while the pink arrows represent the profile of the smaller square pattern. Square patterns have been chosen since they are the easiest to understand and have only two force vectors. As can be seen in the figure, the centres of the forces are not at the same spot, which is assumed to be where the shrinking is centred, thus creating a large strain difference. Figure 2c presents the top view of the model of the actuator, which is made of a bottom layer of small squares and a top layer of larger squares for a variable pattern size sample. This concept of the difference in the strain profile is the driving force for the actuation in this study. Ten actuators are characterized in this section; five are in the larger patterns on the top, and vice-versa.

### 2.4. Mixed Patterns

The final set of actuators was printed with all the variations that are possible by mixing two different patterns into one actuator, each one with its own layer. This section explores the SME of patterns of completely different profiles. This does not necessarily show a stronger SME than the previous action, but it gives a lot of information about the combination of the different stiffness patterns to find certain patterns that act better in certain positions. Since there are five different patterns, by testing all possible combinations, a total of 20 different actuators are needed. The size of the shapes is the same as the shapes of the first set since this section does not aim to maximize the SME but to assist in understanding the difference in the stiffness profiles of each pattern.

### 2.5. Patterns-Driven Gripper

A four-armed gripper is developed for the testing of the patterns as a proof-of-concept design. The gripper uses two different patterns to allow controlling the final deformation of the gripper. The use of the patterns allows the gripper to hold non-uniform objects by having different deformations in the arms. This is a favoured behaviour when the force exerted by the gripper on the held object should be low enough to not damage a delicate object. This characteristic of controllable force in soft grippers gives them an advantage over uniform deformation grippers.

## 3. Fabrication Process and Experimental Setup

The 3D computer-aided design (CAD) models were created using SolidWorks software. The designs consisted of creating the basic shapes and replicating them to reach the desired pattern size. The arrangement of the shapes in the patterns was made to optimize the number of shapes to be used while keeping optimal contact between the shapes to aid the printing. SolidWorks was then used to generate the STL files that the 3D printing software uses, with a total of 40 actuators. The slicer software used in this work was the Simplify3D slicer, which was favourably used because of the degree of control it allows and the lack of automatic software control. In the slicer, all parameters of the print can be controlled, along with simple edits and orientations of the models used.

The actuators were printed using PLA filament as a construction material via FDM printing. All actuators were set to the same printing settings to ensure that the only difference noticed in the SME was caused by the patterns. The printing temperature (nozzle temperature), bed temperature, and print environment temperature were set to 190 °C, 45 °C, and 25 °C, respectively. The layer thickness was set to 0.1 mm, amounting to a total of 20 layers per sample, and the printing speed was 60 mm/s. The slicer then generated a G-code file that the printer could read and follow to guide the printer through all aspects of the print. The printer used in this study is an FDM printer Ender 3 V2 (Creality, Shenzhen Creality 3D Technology Co., Ltd., Shenzhen, China). The printer was a single extruder Cartesian base, glass-bed printer. The thermoplastic material of the filament used was

polylactic acid (PLA 1.75 mm PLA Pro Filament 1.0 kg Gray, Flashforge, Flashforge 3D Technology, Co., Ltd., Jinhua, China). The design, fabrication, and testing are shown graphically in Figure 3a.

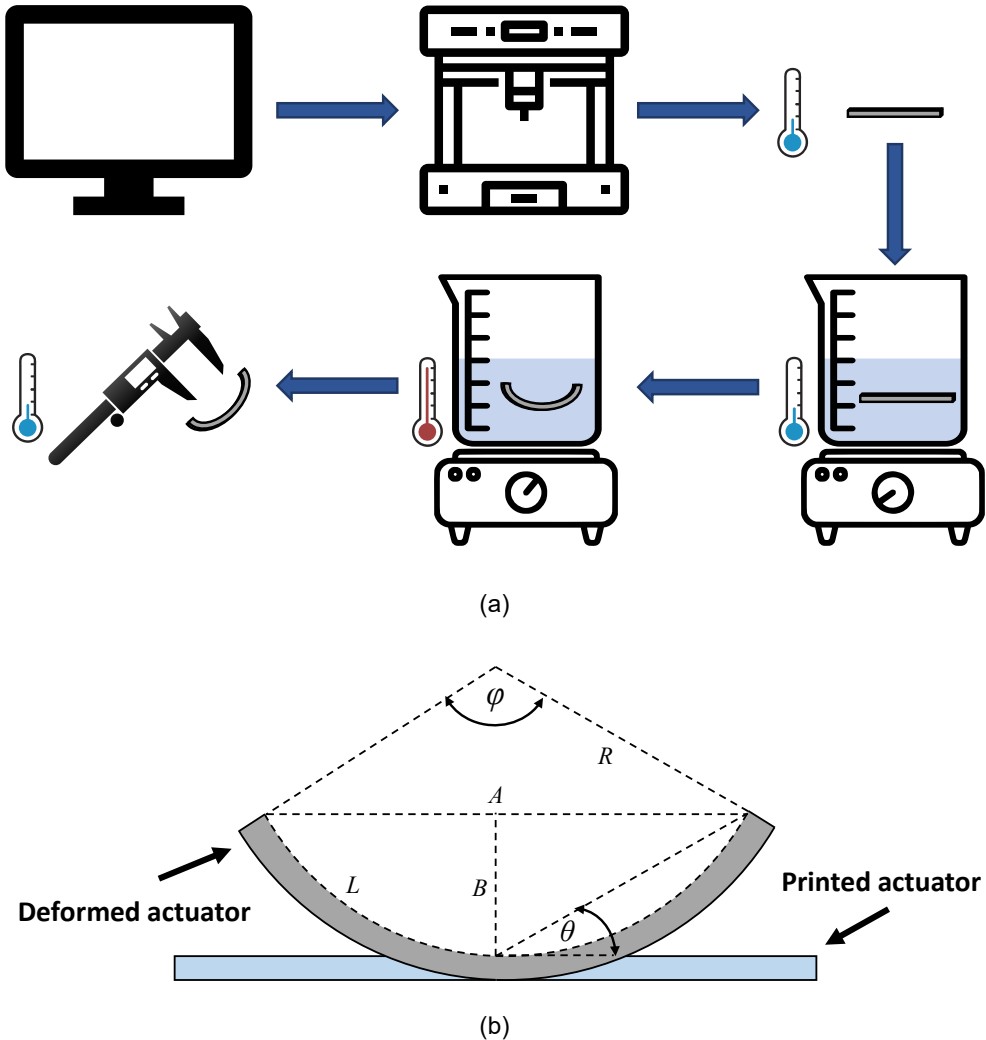

(a)

(b)

**Figure 3.** (**a**) The process of printing and testing the actuators. (**b**) The measurement method of the bending angles in the actuators. *A* and *B* are the measured lengths, while the bending angle is referred to as *θ*. *L* is the arc length measured between the two ends of the actuator.

The testing of the SME of the actuators was done by placing the actuator in a beaker of hot water. The water in the beaker was monitored and constantly heated to keep it at a temperature of about 80 °C, which was higher than the $T_g$ of PLA (~65 °C). This was done to ensure that a full release of the internal strain was accomplished. The dimensions of the actuator were then measured using a digital calliper in order to find the bending angle. The testing of the effect of the internal strain was done by measuring the angle (*θ*) made by the actuators from the centre of the actuator to the endpoint, as shown in Figure 3b, where *A* is the length of the line between the two endpoints of the actuator, and *B* is the length of the line from line *A* to the bottom of the actuator. The angle of bending was found using the following equation:

$$\theta = \arctan\left(\frac{2B}{A}\right) \tag{1}$$

The angle *θ* is used as the measurement for the strength of the internal strain difference, as clearly, the larger angle means there is a larger internal strain difference between the layers. The radius of curvature, which is the radius of the circle formed by the actuators is

represented by $R$, while the length of the arc formed by the actuators after deformation is represented by $L$. The angle that is formed by the arc as a section of the circle is represented by $\varphi$ (measured in radians). Thus, $R$, $\varphi$, and $L$ can be found using Equations (2)–(4):

$$R = \frac{A^2 + 4B^2}{8B} \tag{2}$$

$$\varphi = 2\sin^{-1}\left(\frac{A}{2R}\right) \tag{3}$$

$$L = R \times \varphi \tag{4}$$

## 4. Results and Analysis

The material properties that are tested in this section are those that are needed for material stimuli, mainly the melting temperature ($T_m$) of the polymer and the $T_g$, where the state of the polymer changes from glassy to rubbery, are needed for the actuators. Although PLA is a widely used material with many characterizations having been done, the results are slightly different from one filament to another. The differences in the properties of PLA in the test are due to the nature of the filament since the polymers have slightly different additives and impurities that can change the material properties. Differential scanning calorimetry (DSC) and dynamic mechanical analysis (DMA) tests have been performed on the printed filament to characterize its material properties. The following sections discuss the characterization results of the material properties of PLA, as well as the deformation results of the single pattern, variable pattern size, and different pattern actuators. The testing of the actuators was carried out by printing each sample three times to assess the repeatability of the SME effect. The recorded results are presented by means of average and standard deviation values.

### 4.1. Material Properties

The results from the DSC test are presented in Figure 4a. As is demonstrated in the figure, the DSC was carried out for heating and cooling cycles. From the heating graph, there seem to be two distinct dips in the heat flow, which are marked by the two arrows at temperatures of 53.5 °C and 151.5 °C. The first dip represents the $T_g$ of the material, which is slightly different from the literature $T_g$ of PLA (about 60 °C to 65 °C) [51,52]. The variance might be because of the impurities in the materials, but mainly, it is because $T_g$ is usually recorded using a DMA machine, which measures a slightly different value. This is also shown in the DMA results in Figure 4b, with the peak of the tan $\delta$. The second dip in the DSC test represents $T_m$ of PLA, which is expected to be below 180 °C, based on the literature. As long as the nozzle temperature stays above this point, the print can be done. The heat flow graph seems to have no other significant characteristics along the rest of the temperature range. The same can be seen in the cooling of the materials, as there is no significant characteristic along the curve. There do not seem to be any irregularities in the change in the heat flow, indicating a regular decrease in the heat capacity.

Figure 4b shows the effect of the material temperature over the storage modulus, which reflects the elasticity of the material, and the tan $\delta$. The storage modulus beyond $T_g$ is more than 100 times smaller than that above $T_g$. This reflects the elasticity of PLA at rubbery temperatures, making it not only extremely mendable but with a high capacity for storing shape memory. The point at which the storage modulus starts to stabilize is the same point at the peak of the tan $\delta$ graph, which represents $T_g$. The $T_g$ in the DMA test was found to be around 67.3 °C, which is higher than that measured using the DSC, as expected. From the results, it can be confirmed that the material is to be activated at an environment temperature higher than 67.3 °C; a temperature of 85 °C will be used for the activation.

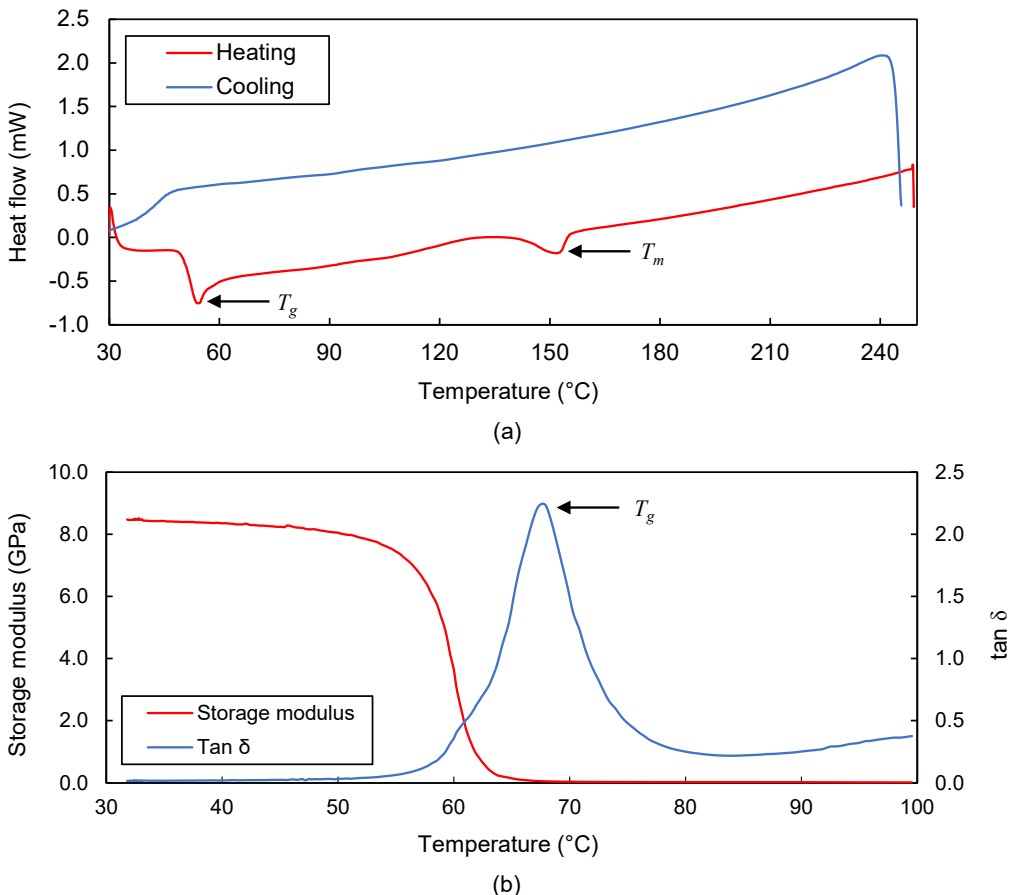

**Figure 4.** Material analysis results. (**a**) DSC results for the heating and cooling of the materials. A small sample of the PLA filament is used in the machine. (**b**) DMA results for a temperature range of 30 °C to 100 °C, rectangular strips of PLA printed with the same settings as the printed actuators are used for the test.

### 4.2. Single Patterns

This section presents the actuation results of the bioinspired single pattern actuators using different dimensions of the aforementioned shapes. The actuation as observed in the results is due to the printing parameters acting on each layer differently, thus creating a small strain difference. The testing of this section is conducted to assess the effect that the shapes have on the actuation of the pattern as a whole. The circular-pattern actuators are presented in Figure 5a for the large circles and (b) for the small circles. The larger shapes have a larger bending than the smaller ones because of the length of the sides of the geometrical shape printed in one movement since larger lengths of lines would have a larger difference in the length after actuation and larger shrinking. Both actuators are bent downwards, implying that if the SME is shrinking, the bottom layers shrink more than the top, causing movement in the downward direction. The squared-pattern actuators are shown in Figure 5c,d, with the large squares in (c) and small ones in (d). The bending direction of both actuators is downward, which is a good indicator of the replicability of the actuation in new prints because the nature of the printing methods causes the bottom layers to shrink larger than the top layers in both circles and squares.

The hexagon-pattern actuators are shown in Figure 5e,f. The uneven shape change on a certain side can be caused by the print itself due to impurities and mixtures. Moreover, the difference in the deformed shape occurs due to the imperfections of the printing process, which require each line of the actuator to be printed perfectly at the same speed and temperature regardless of the environmental effects [27]. In addition, in some cases, the air gets trapped in the prints, whether because of the feeder or bubbles between the layers. The

fourth shape that is used is the rhombus, as shown in Figure 5g,h. The bending behaviour is similar to the previous shapes but shows different bending angles. The last shape used in this study is the triangle, as shown in Figure 5i,j. Although the deformation follows the trend of larger shapes causing a larger SME, in these actuators, the bending angle difference between the larger and smaller triangles is large.

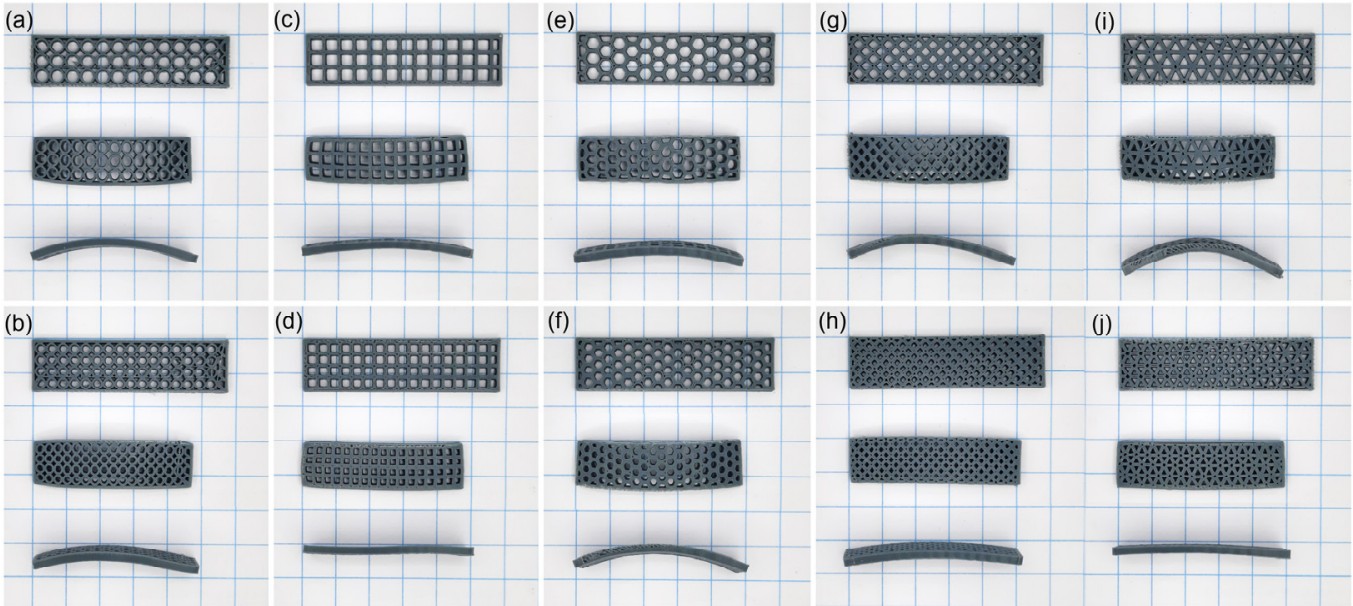

**Figure 5.** Single pattern actuators with each set showing the top view of the actuator before activation, top view of the actuator after activation, and side view of the actuator after activation from top to bottom, respectively, on a 1 cm square mesh. (**a**) Large circles. (**b**) Small circles. (**c**) Large squares. (**d**) Small squares. (**e**) Large hexagons. (**f**) Small hexagons. (**g**) Large rhombuses. (**h**) Small rhombuses. (**i**) Large triangles. (**j**) Small triangles.

The actuation performance of the single pattern actuators tests is summarised in Table 1, where BD represents the bending direction, and $W$ is the weight of the actuator. In the table, the classification of the actuators is made by the top and bottom patterns, and the measurements $A$ and $B$ denote the length from one end of the actuator to the other end and the height of the actuator after deformation, respectively. From the table, it can be observed that all the actuators bend downwards and shrink after deformation. This can be accredited to the print and the fact that all the stiffness profiles of the patterns are the same across the actuators. There seems to be a trend where the larger the shape used, the larger the bending angles are. This holds true for four out of the five patterns, while the exception is found in the tests of the hexagon patterns. From the calculations of the arc length, denoted by $L$ in the table, the actuators shrink as they deform, where the shrinkage is about 12%. The nature of the printing method of these patterns at the specific setting makes the actuators bend downwards, which is seen in all patterns. The largest shape change has been observed in the pattern of the larger triangles, while the smallest bending angle was observed in the squares of smaller shapes. In addition, it was observed that there is no relationship between the weight of the actuators and their deformation, which proves that the deformation value is only related to the printing patterns when printing all actuators using the same settings.

**Table 1.** A summary of the actuation performance of the single pattern actuators.

| No. | Bottom Pattern | Top Pattern | BD | W (g) | A (mm) | B (mm) | L (mm) | R (mm) | θ (°) |
|-----|---------------|-------------|-----|-------|--------|--------|--------|--------|-------|
| 1 | 4 mm circles | 4 mm circles | Down | 0.68 | 41.58 ± 0.33 | 5.95 ± 0.30 | 43.82 ± 0.54 | 39.37 ± 1.16 | 15.96 ± 0.66 |
| 2 | 3 mm circles | 3 mm circles | Down | 0.80 | 42.00 ± 0.15 | 5.14 ± 0.05 | 43.66 ± 0.13 | 45.44 ± 0.63 | 13.76 ± 0.16 |
| 3 | 4 mm squares | 4 mm squares | Down | 0.79 | 43.07 ± 0.04 | 4.27 ± 0.04 | 44.19 ± 0.02 | 56.48 ± 0.58 | 11.21 ± 0.11 |
| 4 | 3 mm squares | 3 mm squares | Down | 0.94 | 43.33 ± 0.25 | 2.35 ± 0.04 | 43.67 ± 0.25 | 101.08 ± 1.58 | 6.19 ± 0.09 |
| 5 | 4 mm hexagons | 4 mm hexagons | Down | 0.79 | 42.36 ± 0.23 | 5.12 ± 0.08 | 43.99 ± 0.22 | 46.38 ± 0.82 | 13.59 ± 0.23 |
| 6 | 3 mm hexagons | 3 mm hexagons | Down | 0.94 | 43.59 ± 0.45 | 6.75 ± 0.18 | 46.33 ± 0.30 | 38.59 ± 1.53 | 17.22 ± 0.6 |
| 7 | 4 mm rhombuses | 4 mm rhombuses | Down | 0.98 | 42.44 ± 0.95 | 6.96 ± 0.69 | 45.44 ± 1.46 | 36.01 ± 1.45 | 18.13 ± 1.31 |
| 8 | 3 mm rhombuses | 3 mm rhombuses | Down | 1.17 | 45.47 ± 0.94 | 7.03 ± 0.40 | 48.33 ± 0.75 | 40.47 ± 2.90 | 17.19 ± 1.14 |
| 9 | 4 mm triangles | 4 mm triangles | Down | 1.02 | 41.43 ± 0.22 | 8.67 ± 0.29 | 46.12 ± 0.43 | 29.11 ± 0.64 | 22.71 ± 0.65 |
| 10 | 3 mm triangles | 3 mm triangles | Down | 1.22 | 44.82 ± 0.16 | 3.36 ± 0.16 | 45.49 ± 0.20 | 76.64 ± 3.21 | 8.52 ± 0.38 |

### 4.3. Variable Pattern Sizes

This section presents the actuation results of the bioinspired actuators printed with two pattern sizes, one at the bottom and one at the top. The first set of actuators is printed with circular-shaped patterns in two different shape sizes, as presented in Figure 6a,b. As is seen in the actuators, the flipping of the pattern sizes has flipped the actuation direction, which is a good indication of the effect on the action each pattern has.

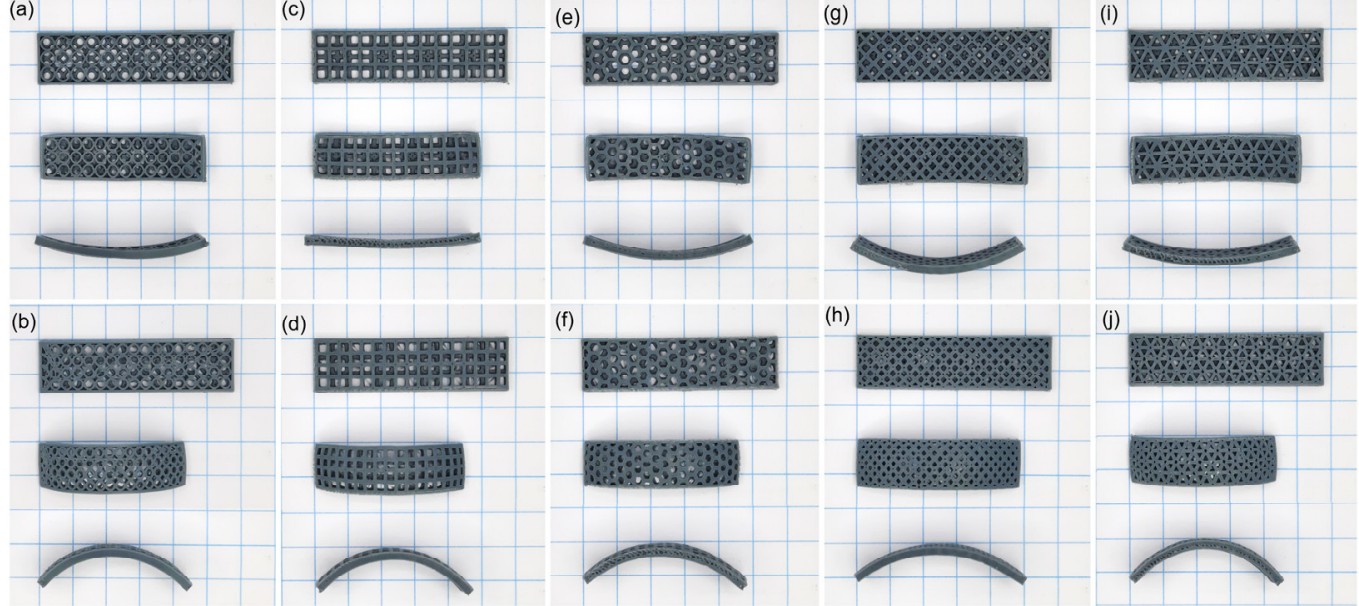

**Figure 6.** Variable pattern size actuators with each set showing the top view of the actuator before activation, top view of the actuator after activation, and side view of the actuator after activation from top to bottom, respectively, on a 1 cm square mesh. (**a**) Large-small circles. (**b**) Small-large circles. (**c**) Large-small squares. (**d**) Small-large squares. (**e**) Large-small hexagons. (**f**) Small-large hexagons. (**g**) Large-small rhombuses. (**h**) Small-large rhombuses. (**i**) Large-small triangles. (**j**) Small-large triangles.

The bending angle of the pattern with the smaller shapes at the bottom is 16.16° upwards, while the reverse has a bending angle of 30.27° downwards, which means the difference is great. The bending angle in the actuator with the smaller patterns on the bottom is higher than all the actuators of a single pattern, meaning a higher strain is present in the border between the two patterns. As was demonstrated, the printing of one size of shape before another does not produce the same results as flipping the actuator. This is because the printing method has the tendency to make the actuators bend downwards, as was explained in the previous section. This effect makes the bending in the upwards direction occur at a smaller angle than that in the downwards direction.

The second tested shapes are the squares, which are presented in Figure 6c,d. The bending in the actuators follows the same trend as the previous shapes, with the actuator

with the smaller shapes bending upwards and the other bending downwards at a much higher angle. The bending angle of the first actuator is 9.77°, and the second is 27.39°. As compared to the actuators with different circle sizes, the square actuators have the same bending angle, which might be some evidence that the amount of strain between the two shape sizes is the same. However, this does not hold when comparing the actuators with the smaller shapes on the bottom since the bending angle is less in the square than it is in the circle. This behaviour shows a direct effect of the use of the shapes and not just the strain difference or the induced stress. Although the SME in the circles and squares is similar and is shown to cause the same bending angle in one of the actuators, the effect of the stiffness profile seems to influence the SME as well.

The third set of actuators includes those using the hexagon shapes, presented in Figure 6e,f. As compared to the hexagon actuators from the previous section, this print has a much higher bending angle. The actuators demonstrated in Figure 6g,h are in rhombus shape patterns. As with the last three actuator patterns, these follow the same bending trend. The bending angles of the two actuators are similar, with values of 18.24° and 25.79°. Since switching the position of the smaller and larger shapes has a similar effect to simply flipping the actuators, it has been demonstrated that the strain or stiffness difference between the two sizes of the shapes is the major contributor to the bending in these patterns.

The last set of variable pattern size actuators in this work is the triangle. The triangle shape pattern actuators are shown in Figure 6i,j. The bending angles of the actuators are 15.04° and 30.86° for the smaller triangles at the bottom and the larger triangles at the bottom, respectively. The triangles provide the best combination of directional force, combined with the large strain difference between the smaller and larger triangles. Table 2 presents a summary of all the variable pattern size actuators in this work, allowing for a general overview of the effect of the shape sizes.

**Table 2.** A summary of the actuation results of the variable pattern size actuators.

| No. | Bottom Pattern | Top Pattern | BD | $W$ (g) | $A$ (mm) | $B$ (mm) | $L$ (mm) | $R$ (mm) | $\theta$ (°) |
|-----|----------------|-------------|-----|---------|----------|----------|----------|----------|--------------|
| 11 | 3 mm circles | 4 mm circles | Up | 0.74 | 42.61 ± 0.91 | 6.17 ± 0.17 | 44.96 ± 0.76 | 39.94 ± 2.36 | 16.16 ± 0.71 |
| 12 | 4 mm circles | 3 mm circles | Down | 0.74 | 38.94 ± 0.40 | 11.37 ± 0.53 | 47.27 ± 0.83 | 22.39 ± 0.61 | 30.27 ± 1.16 |
| 13 | 3 mm squares | 4 mm squares | Up | 0.87 | 44.63 ± 0.86 | 3.84 ± 0.07 | 45.51 ± 0.85 | 66.80 ± 2.30 | 9.77 ± 0.20 |
| 14 | 4 mm squares | 3 mm squares | Down | 0.87 | 40.51 ± 0.10 | 10.51 ± 0.94 | 47.48 ± 1.09 | 24.94 ± 1.48 | 27.39 ± 2.16 |
| 15 | 3 mm hexagons | 4 mm hexagons | Up | 0.87 | 43.04 ± 0.71 | 7.09 ± 0.55 | 46.12 ± 0.32 | 36.48 ± 3.13 | 18.24 ± 1.59 |
| 16 | 4 mm hexagons | 3 mm hexagons | Down | 0.87 | 42.23 ± 0.92 | 10.20 ± 0.10 | 48.52 ± 0.81 | 26.97 ± 0.96 | 25.79 ± 0.52 |
| 17 | 3 mm rhombuses | 4 mm rhombuses | Up | 1.08 | 45.46 ± 0.96 | 8.26 ± 0.30 | 49.37 ± 1.04 | 35.42 ± 1.22 | 19.98 ± 0.59 |
| 18 | 4 mm rhombuses | 3 mm rhombuses | Down | 1.08 | 44.37 ± 0.53 | 9.85 ± 0.45 | 49.99 ± 0.92 | 29.94 ± 0.52 | 23.93 ± 0.77 |
| 19 | 3 mm triangles | 4 mm triangles | Up | 1.12 | 45.11 ± 0.39 | 6.06 ± 0.28 | 47.25 ± 0.22 | 45.12 ± 2.53 | 15.04 ± 0.76 |
| 20 | 4 mm triangles | 3 mm triangles | Down | 1.12 | 41.30 ± 3.29 | 12.30 ± 0.44 | 50.49 ± 3.20 | 23.53 ± 2.44 | 30.86 ± 1.27 |

From the table, it can be noted that when the smaller patterns are printed at the bottom and the larger patterns are printed on top of them, the bending is in the upwards direction. The exact opposite is observed, which indicates the effect of the variable stiffness profile on the bending direction and force since the strain difference is always strong enough to change the direction of the bend. It was also observed that there is no relationship between the weight of the actuators and their deformation, similar to Designs 1–10.

### 4.4. Mixed Patterns

This section presents the actuation results of the mixed bioinspired pattern actuators. The experiments were carried out to investigate the strain difference generated between each of the patterns. Only the large patterns are used so that the effect of the smaller patterns does not influence the bending, as the aim of this section is not to maximize the bending but to explore the effect of the patterns on the SME. A total of 20 actuators with all the possible variations that can be made by combining two different patterns and their reversed order were characterized, as shown in Figure 7. The variable stiffness profiles created by the different combinations are different from those in the previous sections, as they are of different shapes.

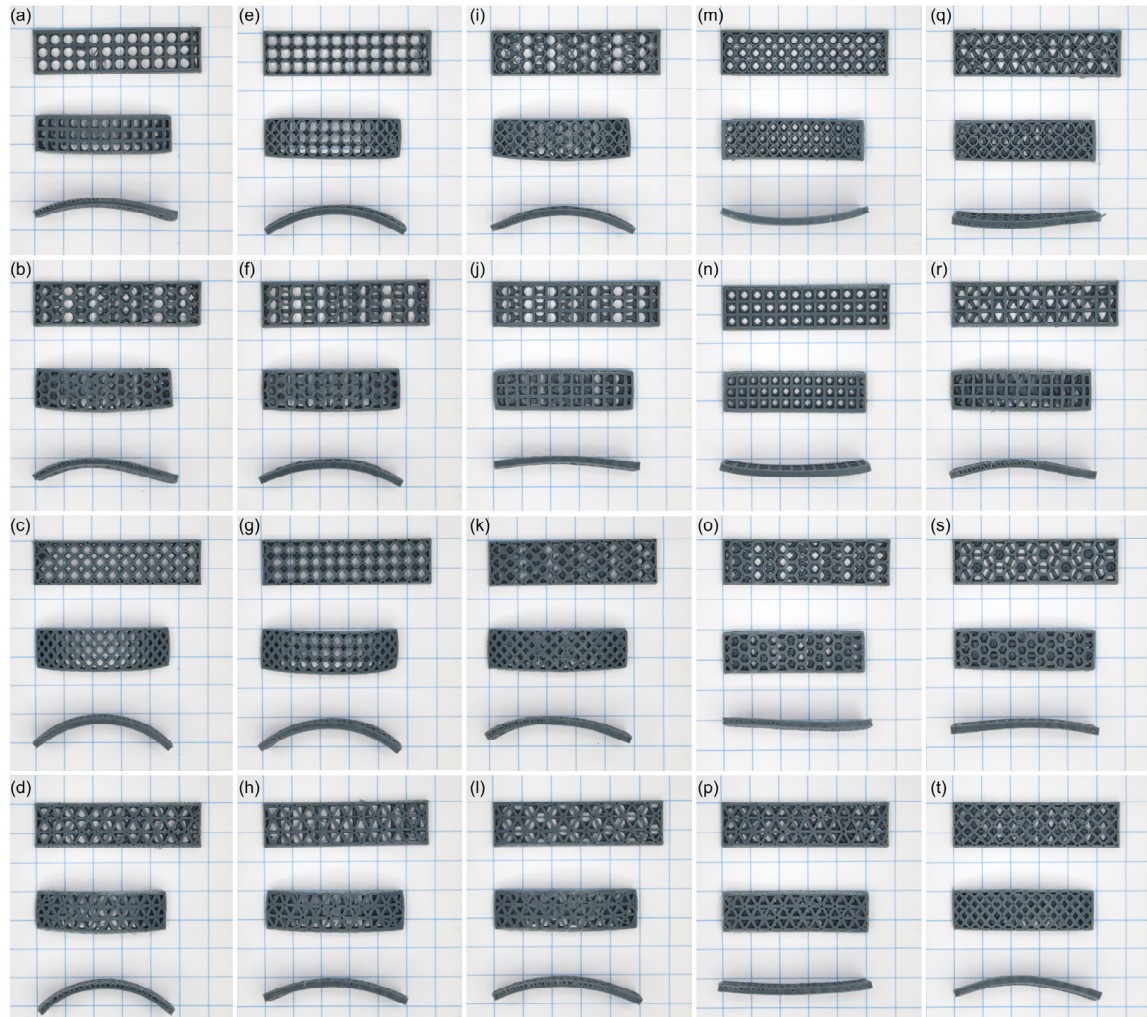

**Figure 7.** Mixed pattern actuators with each set showing the top view of the actuator before activation, top view of the actuator after activation, and side view of the actuator after activation from top to bottom, respectively, on a 1 cm square mesh. (**a**) Circles-squares. (**b**) Circles-hexagons. (**c**) Circles-rhombuses. (**d**) Circles-triangles. (**e**) Squares-circles. (**f**) Squares-hexagons. (**g**) Squares-rhombuses. (**h**) Squares-triangles. (**i**) Hexagons-circles. (**j**) Hexagons-squares. (**k**) Hexagons-rhombuses. (**l**) Hexagons-triangles. (**m**) Rhombuses-circles. (**n**) Rhombuses-squares. (**o**) Rhombuses-hexagons. (**p**) Rhombuses-triangles. (**q**) Triangles-circles. (**r**) Triangles-squares. (**s**) Triangles-hexagons. (**t**) Triangles-rhombuses.

Designs 21–24 consist of all the combinations allowed with the circles at the bottom, as shown in Figure 7a–d, and are identified by the name of the patterns as bottom-top. The patterns used are circles-squares, circles-hexagons, circles-rhombus, and circles-triangles, respectively, where all actuators bent downwards.

Naturally, the direction of bending is downwards unless the strain of the boundary is large enough in the upwards direction to reverse it. A similar approach was used to fabricate Designs 25–28, Designs 29–32, Designs 33–36, and Designs 37–40, where the bottom layers of these groups of designs use squares-patterns, hexagons-patterns, rhombuses-patterns, and triangles-patterns, respectively. Meanwhile, the top layers of these groups of designs use the other patterns that are not used in their bottom layers, as shown in Figure 7e–t. From the figures, it can be seen that most of the actuators bend downwards, except for five designs. Four of these designs are Designs 33–36, which use rhombuses-patterns in

their bottom layers. This can be explained as the result of the effect of the force profiles presented in Figure 1.

Since the rhombus patterns do not have force profiles that are parallel to the lengths of the actuators, then the induced strain values are not maximum along the lengths of the actuators. Thus, these actuators will be more affected by the patterns of their top layers, making them bend upwards. The last actuator that bends upwards is the one that uses the triangles-circles structure (Design 37). The reason behind this behaviour might be caused by the way the patterns of the top and bottom layers align with each other, affecting the overall force profile along the actuator. The effect of the force profiles can also be seen when observing the deformation angles of the designs. A summary of the tested actuators in the final section of this study is presented in Table 3.

**Table 3.** A summary of the actuation results of the mixed patterns actuators.

| No. | Bottom Pattern | Top Pattern | BD | *W* (g) | *A* (mm) | *B* (mm) | *L* (mm) | *R* (mm) | *θ* (°) |
|---|---|---|---|---|---|---|---|---|---|
| 21 | 4 mm circles | 4 mm squares | Down | 0.74 | 42.76 ± 0.57 | 6.47 ± 0.19 | 45.33 ± 0.62 | 38.56 ± 0.96 | 16.85 ± 0.42 |
| 22 | 4 mm circles | 4 mm hexagons | Down | 0.74 | 42.81 ± 0.09 | 6.53 ± 0.08 | 45.41 ± 0.05 | 38.37 ± 0.52 | 16.96 ± 0.23 |
| 23 | 4 mm circles | 4 mm rhombuses | Down | 0.83 | 41.35 ± 0.09 | 9.49 ± 0.78 | 46.97 ± 0.94 | 27.42 ± 1.45 | 24.63 ± 1.76 |
| 24 | 4 mm circles | 4 mm triangles | Down | 0.85 | 41.35 ± 0.43 | 10.17 ± 0.54 | 47.74 ± 0.87 | 26.16 ± 0.77 | 26.17 ± 1.11 |
| 25 | 4 mm squares | 4 mm circles | Down | 0.79 | 42.60 ± 0.26 | 7.23 ± 0.46 | 45.81 ± 0.62 | 35.10 ± 1.51 | 18.74 ± 1.03 |
| 26 | 4 mm squares | 4 mm hexagons | Down | 0.89 | 43.62 ± 1.21 | 9.94 ± 0.49 | 49.44 ± 1.35 | 28.96 ± 1.36 | 24.49 ± 1.01 |
| 27 | 4 mm squares | 4 mm rhombuses | Down | 0.91 | 43.70 ± 0.54 | 7.43 ± 0.15 | 46.99 ± 0.63 | 35.86 ± 0.27 | 18.77 ± 0.16 |
| 28 | 4 mm squares | 4 mm triangles | Down | 0.89 | 43.81 ± 0.98 | 7.89 ± 0.60 | 47.52 ± 1.45 | 34.44 ± 0.70 | 19.78 ± 0.98 |
| 29 | 4 mm hexagons | 4 mm circles | Down | 0.91 | 42.58 ± 0.93 | 6.70 ± 0.28 | 45.34 ± 0.97 | 37.24 ± 1.60 | 17.46 ± 0.66 |
| 30 | 4 mm hexagons | 4 mm squares | Down | 1.00 | 45.57 ± 0.18 | 3.65 ± 0.13 | 46.35 ± 0.14 | 73.06 ± 2.95 | 9.10 ± 0.34 |
| 31 | 4 mm hexagons | 4 mm rhombuses | Down | 0.74 | 42.26 ± 0.55 | 9.58 ± 0.86 | 47.87 ± 1.21 | 28.27 ± 1.68 | 24.36 ± 1.86 |
| 32 | 4 mm hexagons | 4 mm triangles | Down | 0.74 | 43.22 ± 0.95 | 8.19 ± 0.25 | 47.25 ± 1.10 | 32.59 ± 0.53 | 20.76 ± 0.20 |
| 33 | 4 mm rhombuses | 4 mm circles | Up | 0.83 | 44.69 ± 0.65 | 5.68 ± 0.33 | 46.59 ± 0.74 | 46.93 ± 2.10 | 14.25 ± 0.73 |
| 34 | 4 mm rhombuses | 4 mm squares | Up | 0.85 | 43.88 ± 0.62 | 3.33 ± 0.14 | 44.55 ± 0.64 | 73.97 ± 2.59 | 8.64 ± 0.31 |
| 35 | 4 mm rhombuses | 4 mm hexagons | Up | 0.79 | 43.12 ± 1.51 | 4.08 ± 0.41 | 44.15 ± 1.66 | 59.32 ± 2.64 | 10.69 ± 0.73 |
| 36 | 4 mm rhombuses | 4 mm triangles | Up | 0.89 | 44.97 ± 0.21 | 4.04 ± 0.12 | 45.93 ± 0.25 | 64.69 ± 1.44 | 10.18 ± 0.27 |
| 37 | 4 mm triangles | 4 mm circles | Up | 0.91 | 45.14 ± 1.03 | 5.59 ± 0.12 | 46.97 ± 1.06 | 48.34 ± 1.30 | 13.92 ± 0.12 |
| 38 | 4 mm triangles | 4 mm squares | Down | 0.89 | 44.81 ± 0.40 | 3.33 ± 0.10 | 45.47 ± 0.42 | 77.10 ± 2.04 | 8.45 ± 0.23 |
| 39 | 4 mm triangles | 4 mm hexagons | Down | 0.91 | 45.29 ± 0.75 | 3.90 ± 0.66 | 46.2 ± 0.67 | 70.09 ± 14.01 | 9.78 ± 1.69 |
| 40 | 4 mm triangles | 4 mm rhombuses | Down | 1.00 | 44.04 ± 0.53 | 6.86 ± 0.38 | 46.84 ± 0.73 | 38.85 ± 1.42 | 17.30 ± 0.79 |

The bending angels of the mixed pattern actuators are relatively smaller than those of the different shape-sizes actuators and appear to have a weaker strain at the boundary, as indicated by the direction of bending, which is mostly downwards. The shrinkage, in this case, is about the same, at about 10%, with little variance. It can be seen that there is no relationship between the weight of the actuators and their deformation, similar to Designs 1–20.

*4.5. Discussion*

Among all the developed designs, five designs provided notable performance compared to the rest of the designs. These designs were selected because they either provided the best or worst performance criteria among the other designs. For instance, in terms of *θ*, Design 20 was the highest, with a bending angle of 30.86°. This also caused the length of *B* to be the highest since it is proportional to *θ*, while the opposite was observed when it comes to the length of *R* since it is inversely proportional to *θ*. The values of *B* and *R* in Design 20 were 12.30 and 23.53 mm, respectively. The design was printed using the bottom and top layers of 4 mm triangles and 3 mm triangles, respectively. On the other hand, Design 4 showed the poorest performance among the rest of the designs. This design was printed using the bottom and top layers of 3 mm squares. The bending angle was 6.19°, causing the lengths of *B*, *L*, and *R* to be the poorest, at values of 2.35, 43.67, and 101.08 mm, respectively.

Design 30 achieved the highest length value of *A* (45.57 mm). However, this affected its bending performance, making it the third lowest design with 9.10°. Design 30 was printed using 4 mm hexagons and 4 mm squares for the bottom and top patterns, respectively. On the contrary, Design 12 showed the lowest longitudinal length, where the length of *A* was 38.94 mm. However, the resultant banding angle (30.27°) was the second highest bending angle. This design was printed using circle patterns of 4 and 3 mm for the bottom and top

layers, respectively. Design 2 showed the lowest *L* value (43.66 mm) with a poor bending performance of 13.76°. This design was printed using 3 mm circles for the bottom and top patterns. The aforementioned observations are essential when selecting the printing patterns based on the desired performance requirements. A comparison among the selected designs is summarised in Figure 8.

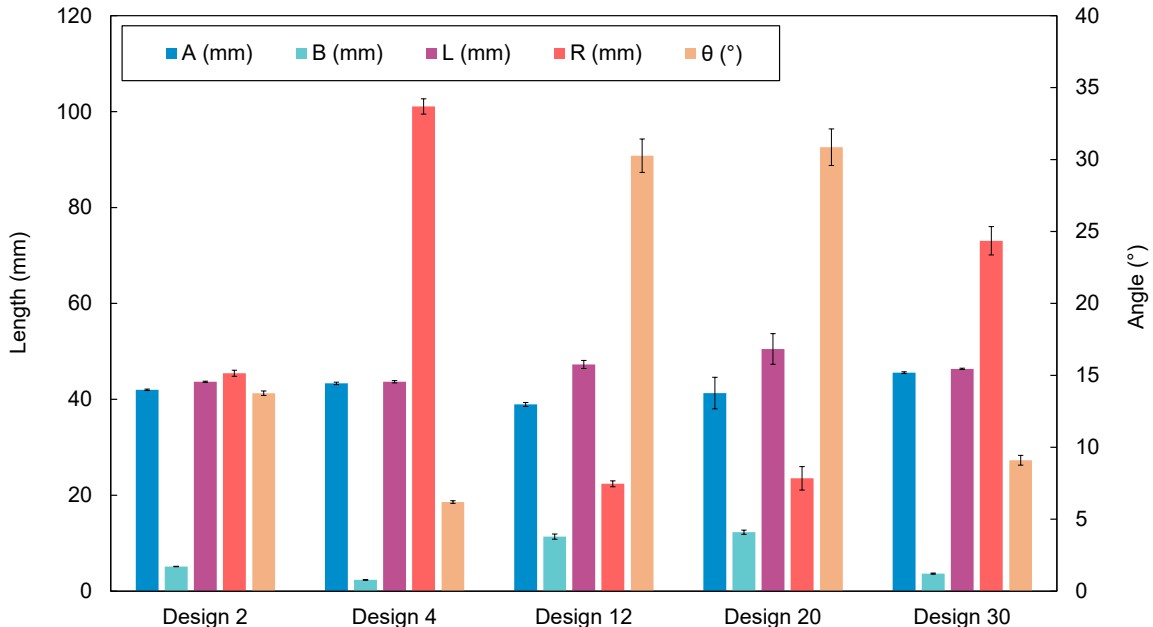

**Figure 8.** Comparison between Designs 4, 12, 18, 20, and 30 based on different performance criteria.

The aforementioned results show that Design 20 achieved ~5 times the bending angle value achieved by Design 4, proving that the actuation range can be varied based on the printing pattern. In addition, it was observed that the average bending angle of Designs 1–10, Designs 11–20, and Designs 21–40 were 15.11°, 21.74°, and 16.61°, respectively. This shows that using various pattern sizes can generally enhance the bending angle when using several printing patterns, as demonstrated previously in Table 2. Moreover, based on the results presented in the previous sections, it can be indicated that the proposed designs demonstrated repeatable results with low standard deviation values, where the average standard deviation values of *A*, *B*, *L*, *R*, and *θ* were 0.63, 0.32, 0.76, 1.56, and 0.71, respectively.

## 5. Proposed Implementation

To demonstrate the feasibility of utilizing the proposed designs in practical applications, a hand-like shaped gripper was developed, as shown in Figure 9. The gripper consists of 4 fingers with a length of 41.60 mm, where Fingers 1 and 3 are based on Design 16 (big and small hexagons), while Fingers 2 and 4 are based on Design 20 (big and small triangles), as shown in Figure 9a. The use of different patterns allows each pair of fingers to achieve different bending angles despite being printed using the same printing parameters. In addition, these patterns allow reducing the weight of the gripper to 3.58 g, making it suitable for mass prototyping of lightweight grippers. To test the gripping performance, the gripper was used to pick objects in two different scenarios. In the first scenario, the gripper was used to pick a rubber cap with a uniform cross-section (circle) at the gripping pints, as shown in Figure 9b–e. After 3 s, the gripper's fingers reached almost half of their final deformation values (Figure 9c), while the complete deformation required around 11 s (Figure 9d). The gripping force was strong enough to hold the 21.15 g cap and lift it outside the water environment even at high temperatures when the gripper was still in a rubbery state. The temperature of the water was 85 °C throughout the whole process. In the

second scenario, the gripper was used to pick an object of a non-uniform cross-section at the gripping points, which is a 49.82 g egg with an oval cross-section and smooth surfaces, making it harder to be picked up. The process started by immersing the gripper in hot water with a temperature of 85 °C to activate the gripper and allow it to enclose the egg. After 3 s (Figure 9g), it can be seen that the triangle patterns are almost touching the surface of the egg, while the hexagon patterns show slower deformation. Then, the gripper was allowed to cool down to a temperature below the start of the transition temperature, which is about 50 °C. The gripper was then in a glassy state and was able to pick up the egg. These steps are presented in Figure 9f–e.

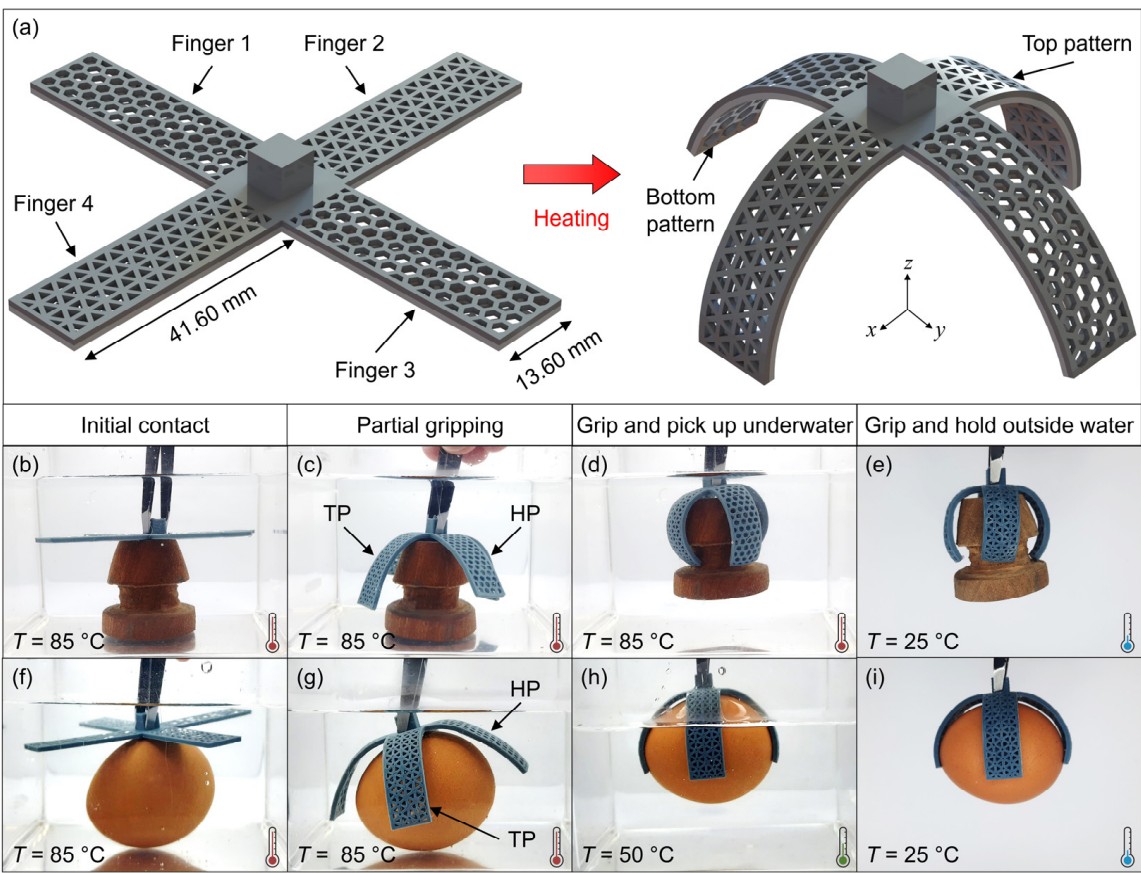

**Figure 9.** Proposed gripper based on the bioinspired patterns. (**a**) The 3D designs of the gripper before and after activation. (**b**–**e**) Different stages of gripping a rubber cap. (**f**–**i**) Different stages of gripping an egg. HP represents the hexagon patterns, while TP represents the triangle patterns. Fingers with triangle patterns deform faster and higher than those with hexagon patterns.

Since the length of the egg is larger than the width, the fingers that grip the width must have a higher bending ratio compared to the ones gripping the length, as can be seen in Figure 9g. After taking the objects out of the water, as shown in Figure 9e,i, it can be seen that the objects were gripped firmly despite the difference in the bending angles of the two pairs of fingers. This is due to the fact that gripping took place during the rubbery state, where slight force values were applied to the objects. Then, at room temperature, PLA was in a glassy state that allowed the fingers of the grippers to hold the objects without damaging them. Such behaviour proves that the proposed gripper is suitable for delicate gripping applications.

The temporal deformation performance of the fingers of the gripper when gripping the rubber cap and egg was observed using Kinovea software. The hexagon patterns (Fingers 1 and 3) deform along the *y*-axis, while the triangle patterns (Fingers 2 and 4)

deform along the *x*-axis, as shown in Figure 10. In both scenarios, it can be noticed that the triangle patterns deform faster than the hexagon patterns, which confirms the observed performance in Figure 9c,g. However, when gripping the rubber cap, the hexagon patterns reached a steady-state value that is equal to the one achieved by the triangle patterns. On the other hand, when gripping the egg, the deformation of the hexagon patterns was lower than that of the triangle patterns since both deformations are restricted by the shape of the egg. This also explains the shorter settling time when gripping the egg.

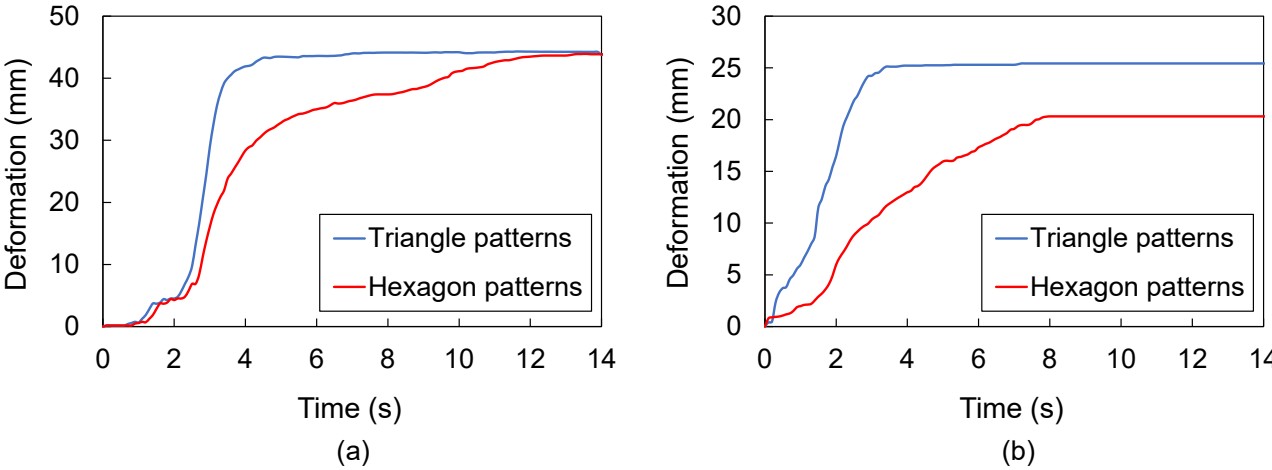

**Figure 10.** Temporal deformation of the fingers of the gripper at 85 °C when gripping: (**a**) the rubber cap and (**b**) the egg. The hexagon patterns deform along the *y*-axis, while the triangle patterns deform along the *x*-axis.

The results show that the proposed approach provides an additional ability to control the gripping performance based on the printing patterns. Thus far, most 4D-printed grippers rely on adjusting the printing speed, printing direction, and printing temperature to achieve gripping [10,30–32]. The use of different printing patterns allows for achieving different levels of bending that can be tailored based on the application. Moreover, this approach allows printing grippers with fingers that include different segments, where each segment consists of different combinations of patterns. Thus, it is possible to print grippers with different bending and stiffness levels at each segment.

## 6. Conclusions

The principle of using the printing parameters to induce internal strain into the actuators in 4D printing can be a useful tool not only for eliminating the need for manual programming but also for having a precise SME effect in the actuators. The use of patterns of variable stiffness can be used to control the degree of bending in the actuators. The use of patterns with shapes of different sizes causes the internal strain between the layers to generate large deformations in the actuators, and it is controllable to a very high degree in terms of the degree and direction of bending. A total of 5 bioinspired shapes were tested with their variations for a total of 40 different designs, of which some were of different pattern sizes, and some were mixed patterns. The shapes used were circles, squares, hexagons, rhombuses, and triangles. The effect of the patterns on the bending performance can be summarized as follows:

- The triangle patterns with different sizes in each layer had the greatest deformation angle, which was 30.86°;
- The rhombus shape patterns seem to reduce the effects of printing parameters on the direction of the bending, allowing the most control;
- Shrinking in all the actuators ranged from 8% to 12%, as measured using the arc length;
- Changing the size of the shapes used was highly useful in determining the direction of the bending in the actuators.

These patterns were also used to design a gripper with fingers that bend at different rates and angles, allowing it to pick up uneven objects. More research can be done on pattern variations and their effects, as well as integrating the patterns into larger, more complex structures. Testing on the force caused by the shape change and thermomechanical characterizations of the prints can be done to enhance the understanding of the SME of the patterns in practical applications.

**Supplementary Materials:** The following are available online at https://www.mdpi.com/article/10.3390/su141610141/s1, Figure S1: Designs dimensions.

**Author Contributions:** Conceptualization, Y.S.A. and M.N.; methodology, Y.S.A., K.B.M., A.Z., M.B., M.S.M.A., H.A.A. and M.N.; validation, Y.S.A. and M.N.; formal analysis, Y.S.A., K.B.M., A.Z., M.B. and M.N.; investigation, Y.S.A. and M.N.; resources, Y.S.A. and M.N.; data curation, Y.S.A. and M.N.; writing—original draft preparation, Y.S.A.; writing—review and editing, K.B.M., A.Z., M.B., M.S.M.A., H.A.A. and M.N.; software, Y.S.A. and M.N.; visualization, M.N.; supervision, M.N., H.A.A. and M.S.M.A.; project administration, M.N.; funding acquisition, M.N. All authors have read and agreed to the published version of the manuscript.

**Funding:** This work was supported by the Ministry of Higher Education Malaysia under the Fundamental Research Grant Scheme (FRGS/1/2019/TK05/UNIM/02/2) and the University of Nottingham Malaysia.

**Institutional Review Board Statement:** Not applicable.

**Informed Consent Statement:** Not applicable.

**Data Availability Statement:** The data that support the findings of this study are available from the corresponding author, M.N., upon reasonable request.

**Conflicts of Interest:** The authors declare no conflict of interest.

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
