# Peer review of "Bioinspired Pattern-Driven Single-Material 4D Printing for Self-Morphing Actuators"

_sustainability, doi:10.3390/su141610141_

Round 1

Reviewer 1 Report

1.The authors discussed SMPs and bioinspired 4D printing, however, the discussions in the introduction about the structure designing and different grippers are limited. Please add more discussions about the structure designing. You can refer to the latest papers such as:

“Bio-Inspired Soft Grippers Based on Impactive Gripping”  https://doi.org/10.1002/advs.202002017

“Bistable and Multistable Actuators for Soft Robots: Structures, Materials, and Functionalities” https://doi.org/10.1002/adma.202110384

“Materials, Actuators, and Sensors for Soft Bioinspired Robots” https://doi.org/10.1002/adma.202003139

2. Eq. 1 cannot show correctly. In addition, Eq. 2 – 4 are used to calculate the angle θ. Although the strain and stress are influenced by θ and L, however, they are also influenced by the sample thickness, which is not discussed in these equations.

3. “The first dip represents the Tg of the material, which is slightly different from the literature Tg of PLA (about 60 °C to 65 °C). The variance might be because of the impurities in the materials, but mainly, it is because Tg is usually recorded using a DMA machine, which measures a slightly different value.”

Please clearly refer the literature which is compared with.

4. There are no scale bars in Fig 5 and 6, or explanations of the mesh size of the background paper. Additionally, bottom views are suggested for better understanding of Fig 6.

5. “Error! Reference source not found. presents a summary of all the variable pattern size actuators in this work, allowing for a general overview of the effect of the shape sizes.” Please double check this typo “Error! Reference source not found.”

6. The authors compared different patterns to have the largest θ, however, they did not compare other parameters, such as weights or theoretical bending rigidities etc.

7. How can the gripper release the objects after gripping? What are the response times for the grippers at different temperatures?

Reviewer 2 Report

The authors designed several different shape memory structural units and fabricated 40 actuators using different combinations of sizes and shapes through additive manufacturing. Then they developed a gripper to demonstrate the application. The design is clever but lacks sufficient novelty and does not solve the practical application problem either. More issues are listed as below:

1.        Please check your writing, especially the method of inserting references and formulas. Also inverse trigonometric functions should be used in “arc + function” format.

2.        What is the reason for choosing heat as a driver compared to magnetic, optical, or electrical drivers? What are the advantages?

3.        On page 3, lines 118 to 129 it shows what have been done, but there is no description of innovative points or highlights?

4.        Line 122 it writes “a novel approach to assessing the effects of geometrical shape patterns”, it should be specified, what is the novel approach?

5.        In section 3.1, I do not think it is bioinspired shapes. Shapes in Figure 1 just happen to be the same shape as these five natural examples, and also these five natural examples presenting this shape have no relevance to the properties discussed in this paper.

6.        I don't get Figure 1(p-t). The pre-stress needs to be applied? If yes, please perform the force analysis directly on a single shaped unit (e.g. drawing on the basis of Figure 1(n), if not please elaborate. Also, is the force profile of a circle correct? Please also indicate clearly.

7.        Scale Bar should be added in Figure 5, Figure 6 and Figure 7.

8.        From Figure 5(g), the bend is not in the center, is it also due to impurities or air bubbles? This is additive manufacturing, so how about the reproducibility of the experiment?

9.        The table is suggested to be replaced by figure to be more illustrative to readers, such as using Figure 3(a), filling in the specific parameters and placing them in the line below the side view in Figure 5.

10.    Since it is a shape memory material, the recovery of the gripper should be shown in Figure 9. In addition, the deformation process should be given a timeline.

In conclusion, there are too many issues for this manuscript.

Reviewer 3 Report

The authors explored the effects of different internal printing patterns in the fabrication of PLA plates able to bend with the temperature, as consequence of the release of internal stress introduced in the structure during the printing phase. The authors identify 5 bioinspired basic patterns that were mixed together, obtaining up to 40 different designs, in order to maximize the bending angle. This proposed research is highly interesting but still incomplete in different aspects, such as the data display and the discussion section. Moreover, a mathematical model able to explain the SME of the plates should be included, as well as a computational model able to predict it.  

Major revision

*Introduction: a more detail definition of 4D printing must be provided, as well as a description of its advantages. Please take as references recent and detailed reviews (e.g, DOI: 10.1039/d1tb01335a - https://doi.org/10.1002/adfm.201805290)

*Some examples and advantages of 4D printed actuators should be included in the introduction section.

*Line 192-197: please provide a comprehensive schematic image that include all the tested configuration and their dimension. This image can be also used in the following paragraph to guide the reader to understand the geometry and dimensions of the described patterns  

*Please add a paragraph in the method section in which the tests performed on the printed filaments are described (lines 308-3021), and a paragraph that describes the complex gripper design, fabrication and testing. It is not correct to include a test in the results section that has not been introduced and described in the method section. 

*The actuation performance reported in Table 1-2-3 are reported as absolute value. Is it an average of multiple tests? Please provide the value as mean ± std and the number of performed tests. In addition, please perform statistical analysis on those data to compare the performances between different shape patterns and complexity.

*Figure 8: please provide the standard deviation of the reported data and performed statistics to highlight potential differences between samples

*Paragraph 4.3: the discussion should be in the end of the manuscript (before the conclusion) and must provide an explanation of the observed phenomena, as well as a comparison with the literature

*A deep discussion and explanation of the obtained movements is missing. The authors should try to better explain the reason behind the differences in movements that occurred between different shapes/geometries. In this section, a mathematical model able to describe the phenomenon should be helpful.

*The reason behind the choose of defined design in the gripper manufacturing must be discussed in detailed, starting from the results in Table 1-2-3. Moreover, a comparison with similar 4D printed actuators that exploit SMPs literature must be included (e.g, Wang, Wei, et al. "Soft grasping mechanisms composed of shape memory polymer based self-bending units." Composites Part B: Engineering 164 (2019): 198-204.)

Minor Revision

*Line 88-105: please add those works in the description of bioinspired 4D printing: DOI: 10.1002/adfm.202105665 - doi: 10.1038/nmat4544.)

*Line 132: please indicate the 3D printing technologies that has been exploited is FDM.

*Line 118 – 121: please ref to Figure 1 in this section

*Line 299 – please define the Tm and Tg

*Figure 5-6-7: please add a scale

*Line 359: “because of the length of the parameters printed in one movement”, please specify which are the parameter you refer to

*Line 377: the impurities the authors refer to should be better investigate, since they interfere with the SME of the structure

 Typos

* Eq 1 is not readable in the file

*Line 386, line 450: “Error! Reference source not found” Please fix this reference

Reviewer 4 Report

The work “Bioinspired Pattern-Driven Single Material 4D Printing for Self-Morphing Actuators” is very interesting. However, the authors have to improve the presentation of it. It is recommended to review the following recommendations and comments.

• The information is very interesting and quite good, however, as the writing progresseschanges style and loses strength in the results and analysis (review writing styles, keep it the same). 6 writers are named, they unify style.

• The numbering sequence is incorrect, section 2 is marked and after subsections 3.1, 3.1, 3.2 and 3.3.

• Between lines 199 to 208, the dimensions of the specimens and the perimeter of the actuator are marked, the thickness of the perimeters of the figures is not indicated (it can be supported by diagrams, plans or improve the wording, since when adding distances they do not coincide with final dimensions). The same in the other two patterns.

• In line 256 you indicate the manufacture of 40 actuators, how many times was each actuator repeated? Is only one print per configuration correct or valid to give a specific result?

• The discussion of what is exposed in the work, which standard is made together with the results, constitutes an essential part in which the results shown in the article must be analyzed, discuss their meaning, analyze their scope, and compare them with others. literature results. It is not enough to say, for example: "as can be seen in figure XX, as..., the... decreases or increases, or in figure XX similar behavior is observed of... with temperature". What needs to be done is to explain if the observed behavior is acceptable, it is what is expected, it agrees with the results of others, if it is similar, the reason for the observed behavior must be explained. After the Introduction section, this section is the one that should have enough citations to justify and convince the reader of the veracity of the results.

• In the results, how many times were the samples tested, only once? Is there repeatability?

• Your results show: “Error! Reference source not found.” Lines 386, 450, 500.

• Figure 5.d is observed wavy, argues the reasons with bibliographical references.

• In line 387 “At the table ??”

• In tables 1, 2 and 3, what is the reason for not placing left theta and right theta, since several pieces have different angles (is there repeatability?) What are the reasons why the angles are not equal? only the impurities? Discuss and cite bibliography.

• 4.3 is repeated on lines 402, 458 and 506 (there are 6 writers, editors and reviewers).

• Between lines 47 to 71, the wording of several sentences is very similar in https://www.sciencedirect.com/science/article/abs/pii/S0014305721004420?via%3Dihub

• Fill in missing citations in various sentences.

• The funds are from 2019, what is the reason for wanting to publish until 2022?

Round 2

Reviewer 3 Report

Most of the revisions have been made as requested. However, few more changes regarding previously reported points (listed below again for reference) should be made:

Comment #3

Old Comments: Line 192-197: please provide a comprehensive schematic image that include all the tested configuration and their dimension. This image can be also used in the following paragraph to guide the reader to understand the geometry and dimensions of the described patterns

Authors’ Response: Thanks for the comment. We apologize that we cannot include all the possible combinations that are tested in the paper. Since the number of tested designs is 40, then including all of them will greatly increase the length of the paper, which is 22 pages at this point. In addition, we hope that Figure 1 clearly shows all the patterns. Thus, their combinations should be just a matter of including each pattern in the top and bottom parts of the actuator, as shown in Figure 2. In addition, the dimensions of all actuators are identical (51.6 mm × 13.6 mm × 2 mm), which were indicated in Lines 239 – 246. We hope that this approach is accepted by the reviewer.

New comment: Please add a schematic image for each configuration in the supplementary materials, thus helping the reader to visualize the exploited geometries without burdening the manuscript length

Comment #6

Old Comments: Figure 8: please provide the standard deviation of the reported data and performed statistics to highlight potential differences between samples

Authors’ Response: Thanks for the comment. The values have been included in Table 1 – 3, based on the previous comment.

New Comment: in the new version of the manuscript the graph does not have error bars, please add them for each data set. Moreover, please perform some statistical analysis between the designs for each geometrical value, in order to highlight potential differences between the design and identified the most performing one(s).

Reviewer 4 Report

The work “Bioinspired Pattern-Driven Single Material 4D Printing for Self-Morphing Actuators” is very interesting. Please note that some changes have been made, however it is recommended that you review the following recommendations and comments.

·       Comment again, that as the writing progresses the writing style changes. Check again.

·       Indicate in writing that each design was manufactured 3 times in order to obtain the average and standard deviation values. L and R did not present a standard deviation.

·       Understanding that the comparisons with the actuators and caliper that you report are limited. However, arguing with physical and mechanical properties of materials would help strengthen their results.

·       Understanding that each design was manufactured 3 times. However, my question is also is there repeatability in each design? Was only one test performed per sample?
